# Scalable Production of High-Quality Silver Nanowires via Continuous-Flow Droplet Synthesis

**DOI:** 10.3390/nano12061018

**Published:** 2022-03-21

**Authors:** Jianming Yu, Lijie Yang, Jing Jiang, Xunyi Dong, Zhiyang Cui, Chao Wang, Zhenda Lu

**Affiliations:** Department of Materials Science and Engineering, College of Engineering and Applied Sciences, Nanjing University, Nanjing 210093, China; jming_yu@163.com (J.Y.); mg1934028@smail.nju.edu.cn (L.Y.); jj27336320382022@163.com (J.J.); dongxunyi@163.com (X.D.); 15905174236@163.com (Z.C.); wangchao@yzu.edu.cn (C.W.)

**Keywords:** continuous-flow synthesis, silver nanowires, flow chemistry, pilot-scale synthesis

## Abstract

Silver nanowires (Ag NWs) have shown great potential in next-generation flexible displays, due to their superior electronic, optical, and mechanical properties. However, as with most nanomaterials, a limited production capacity and poor reproduction quality, based on the batch reaction, largely hinder their application. Here, we applied continuous-flow synthesis for the scalable and high-quality production of Ag NWs, and built a pilot-scale line for kilogram-level per day production. In addition, we found that trace quantities of water could generate sufficient vapor as a spacer under high temperature to efficiently prevent the back-flow or mixed-flow of the reaction solution. With an optimized synthetic formula, a mass production of pure Ag NWs of 36.5 g/h was achieved by a multiple-channel, continuous-flow reactor.

## 1. Introduction

Nanomaterials, as an emerging alternative to traditional bulk materials, have shown great potential in multiple applications, such as catalysis, energy storage and conversion, electronics, and so on [1,2,3,4]. The functional properties of nanomaterials, such as electrical, optical, mechanical, thermal, and reactive behaviors, are closely correlated with their intrinsic physicochemical features and synthesis strategies [5,6,7]. Typically, the high purity and uniformity of nanomaterials are essential to achieve homogenous activity as well as reliable performance. Wet chemistry synthesis is a facile and effective strategy to obtain uniform nanomaterials such as colloidal nanocrystals, and normally, this synthesis occurs as a batch reaction in a closed reactor system [8,9]. A batch reaction affords multiple advantages, for example, relatively simple production equipment, precise synthesis, and easy parameter optimization. Nonetheless, the production capacity of this batch reaction is limited by the scale of the reactor [10,11,12]. Even worse, the uniformity of different batches is difficult to guarantee. Developing a more efficient synthetic approach to realize scale-up high-quality production in a solution route is, therefore, highly desirable [13].

Flow chemistry based on a microreaction process, which improves the mixing of precursors, heat exchange, as well as control of reaction time, has been demonstrated as a powerful alternative to the batch reaction for achieving the scale-up and high-quality production of nanocrystals [8]. Fischer et al. reported the flow synthesis of CdS QDs using a four-way connector as a micromixer for a column reaction tube, in which the solution was rapidly heated or cooled when flowed through the tube, ensuring the precise control of the nucleation and growth of the QDs [14]. Other nanocrystals have been successfully prepared by flow chemistry [15,16,17], in that the small volume of the flow microreaction could improve reactant mixing in both speed and homogeneity, as well as the heating process, leading to a high quality of continuous production [18,19].

Although the flow microreaction technology resulted in the production of a series of nanomaterials, most of these were still limited in the milligram-scale for lab research [20,21,22,23,24,25,26]. To our knowledge, there are very few reports on the flow-chemistry synthesis of Ag NWs in mass production (i.e., kilogram-scale) [27,28,29]. In fact, compared to batch reaction, flow microreaction makes it relatively easy to increase production capacity via proportionately scaling-up the synthetic reactor or the system. However, with an expanded volume, challenges of achieving high quality and uniformity are still arising due to the uneven thermal and compositional distribution along the flow and radial direction.

As an improvement, continuous-flow droplet reactors using an immiscible carrier phase to separate and wrap the droplets in a fluidic channel were demonstrated, which could provide reasonably fast and uniform mass/heat transfer in an individual droplet [30]. However, the dilemma is that additional separation and purification processes are required to remove the carrier phase (usually a liquid such as silicone oil) from the final mixture [31,32].

In this work, we report a novel, carrier-free, continuous-flow droplet reactor using deionized (DI) water vapor as segmentation for the scalable production of silver nanowires (NWs). The droplet reactor is realized by traces of water in ethylene glycol (EG) solvent, which provide multiple advantages, including: (1) trace water (instead of immiscible organic-phase liquid) could form water vapor to segment the reaction phase to form discrete droplets at a high temperature (160 °C); (2) the water vapor was re-dissolved in EG at room temperature, and no secondary separation was required; and (3) the water amount was trace, which negligibly affected the product purity. As a result, pure Ag NWs were obtained by the continuous-flow droplet method using a single-channel (PTFE pipe, inner diameter of 6 mm and length of 2 m). After that, an industrial pilot test system was designed and built with a multi-channel reactor with the same inner diameter, delivering a production capacity of 36.5 g/h of pure Ag NWs.

## 2. Experimental Section

*Materials.* Ethylene glycol (EG, >99%, Aladdin, Shanghai, China), silver nitrate (AgNO_3,_ AR, Aladdin, Shanghai, China), poly(vinylpyrrolidone) (PVP, *Mw* = 58,000, Aladdin, Shanghai, China), CuCl_2_·2H_2_O (AR, Aladdin, Shanghai, China). All chemicals were used as received without further purification. Deionized water (DI water, 18.25 MΩ cm^−1^) was obtained with ultra-pure water equipment. The EG was heated in a vacuum at 120 °C before use to remove traces of water initially present in the solvent.

*Methods.* Polyol-method synthesis of Ag NWs by a single-channel continuous-flow droplet reactor system is shown in Figure 1a. The channel used was PTFE pipes with inner diameters of 1, 3, and 6 mm, respectively; the heating zone was placed in an oil bath filled with simethicone. To prepare precursor A, 0.45 g of PVP was dissolved in 30 mL of EG, and 960 μL of CuCl_2_ EG solution (4 mM) was added after complete dissolution. To prepare precursor B, 0.3 g of AgNO_3_ was added to 30 mL of EG, and then completely dissolved. To simplify the introduction of the carrier phase (water vapor), a trace of DI water (0.3%, *V/V*%) was added into precursor A and B solutions to yield a homogenous water concentration.

The reaction residence time in the heating zone was controlled by tuning the advancement speed of the precursor solution (namely, various flow rate), and an ice water bath was used to cool down the high-temperature mixture in the product collection zone. 

The industrialized pilot test production of Ag NWs was designed according to the experimental parameters determined by the laboratory test. Keeping the maximum pipe inner diameter at 6 mm, a multi-channel parallel was employed to expand the production scale. The chemical raw materials were scaled-up quantities of the above formula.

*Characterization*. The morphology of Ag NWs was studied with a field emission scanning electron microscope (FESEM, Ultra 55, Zeiss, Germany). The calculations of the distribution and average of length and diameter were based on the nanowires (>100) observed in the SEM image.

## 3. Results and Discussion

The synthetic strategy for Ag NWs is based on the typical polyol method in a batch reaction [33,34,35]. Here, EG was employed as both the solvent and reducing agent to reduce AgNO_3_ into metallic Ag NWs. Poly(vinylpyrrolidone) (PVP) was used as the capping agent or surfactant, and a small amount of metal halogen salts (CuCl_2_) was introduced to mediate the growth [36]. For the flow chemistry synthesis attempt, a single-channel continuous-flow reaction system was built (Figure 1a). Two EG solutions of surfactant PVP with trace amounts of CuCl_2_ and precursor AgNO_3_ were prepared first, and then fed into the syringe. After that, they were mixed in the mixing zone at a speed of 11 µL/min, and then flowed through the heating zone: a Teflon pipe with a 2 m length and 1 mm inner diameter (1D) immersed in a 160 °C oil bath. The reaction time was determined by the flow speed. The final product was collected in an ice water bath for later separation. Figure 1b shows the representative scanning electron microscopy (SEM) image of the obtained Ag NWs, in which notable, one-dimensional, large-aspect–ratio nanowires of a one-hundred-micron-scale length were observed. The flow microreaction system could continuously produce high-quality Ag NWs with a production capacity of 1.32 mL/h dispersion and of about 4.2 mg/h pure Ag NWs.

The continuous-flow droplet reactor affords the benefit of quickly screening the reaction conditions. Therefore, the optimized synthetic parameters, such as channel diameter, influence of carrier phase, and flow rate, were investigated. First, to expand the production capacity of Ag NWs, the inner diameter of PTFE pipe was gradually increased from 1 mm to 6 mm, maintaining the reaction time of 70 min. As shown in Figure 2a, when the inner diameter increased to 3 mm, the resultant Ag NWs exhibited a relatively low aspect ratio and uniformity, and a small number of particles was observed. As the pipe diameter increased to 6 mm, the uniformity of the Ag NWs was even worse, and a large number of Ag particles was formed in the SEM image (Figure 2b). These observations are consistent with existing knowledge that an increase in reaction volume would cause the back-flow or mixed-flow of the reaction solution in the heating zone, affecting the nucleation and growth processes, thereby leading to a decline in production quality [30,37].

Thus, it is necessary to introduce a carrier phase to form a droplet microreactor. Here, we introduced trace water into the solvent system to modulate the flow reaction. Typically, a trace of water (0.3%, *V/V*%) in the solution was converted into vapor gas with a huge volume expansion at a high temperature of 160 °C, which could act as the carrier phase to enable segmentation of the continuous solution, leaving the liquid phase domains as independent droplet reactors to avoid turbulence (Figure 2c). As a result, high-quality Ag NWs with significantly improved uniformity were obtained (Figure 2d). Interestingly, the trace amount of water in the synthetic solution caused very little disturbance to the formation of the Ag NWs, and it could be re-merged into the EG dispersion in the collection zone, thus being free of any separation concerns after product purification. Therefore, it is a so-called carrier-free continuous-flow droplet reactor.

Based on the production of Ag NWs using the 6 mm pipe and a trace of water, the influence of flow rate on the morphology control of Ag NWs was evaluated. When the flow rate was controlled at 1800 μL/min, the retention time of polyol reaction in the heating zone was 10 min. As shown in Figure 3a, plenty of Ag nanoparticles and short nanorods were formed at such a fast flow rate that there was insufficient time for the oriented growth of Ag seeds. By slowing down the flow rate to 600 μL/min (retention time 30 min), a mixture of the majority of Ag NWs with an average diameter of 61 nm and a few Ag nanoparticles was formed (Figure 3b). As shown in Figure 3c, when the flow rate reduced to 300 μL/min (retention time 60 min), Ag NWs with an average diameter of 117 nm were achieved with very few Ag nanoparticles. When the flow rate decreased even further (200 μL/min, retention time 90 min), the majority of products maintained the Ag NWs, but the average diameter became wider (Figure 3d). The related effects of flow rates on the morphology of Ag NWs were statistically analyzed and are summarized in Table 1; more than 100 NWs were counted for each sample. It was demonstrated that 60 min of the retention time can guarantee the purity of the Ag NWs, and the long time could lead to the thick diameter and low aspect ratio of the Ag NWs. Therefore, we selected 300 μL/min as the optimal flow rate (retention time of 60 min) for a further scale-up attempt, achieving a high purity along with a high aspect ratio of Ag NWs.

An optimized growth formula for the scalable high-quality production of large-aspect-ratio Ag NWs was obtained, in which a large inner diameter of 6 mm for the PTFE pipe channel (the largest reported, to our knowledge) [21,25,26,30], a trace of water (~0.3% *V/V*%), and a proper flow rate of 300 μL/min (retention time 60 min in the heating zone) were determined for this single-channel, carrier-free, continuous-flow droplet reactor. Then, based on these parameters, we designed a multiple-channel pilot production line for a mass production attempt. As depicted in Figure 4a, two precursor storage tanks (15 L) were connected to feeding pumps for the homogenous mixing of raw materials. After being fully mixed, the precursor solution was evenly flowed into four pipelines with an inner diameter of 6 mm, and then through the heating reactor. Finally, the product dispersion was generated and collected after cooling. The entire process of pumps, valves, heating, and water cooling was automatically controlled by computer programs. The length of each pipe used in the heating zone was 55 m in the pilot test, and the flow rate could reach 60 mL/min to achieve a heating retention time of 60 min. The whole equipment setup is shown in Figure 4b, exhibiting a high level of integration for industrial manufacturing. Figure 4c shows the photo of the final product of Ag NWs dispersion in a large storage tank, and Figure 4d confirms the good uniformity and high aspect ratio of Ag NWs produced in the pilot production. The production capacity of dried Ag NWs was 36.5 g/h, with the yield above 98%. Specifically, the production capacity of Ag NWs by this continuous-flow droplet synthesis can reach the kg scale, as expected. Moreover, it is relatively easy to further increase the production capacity by increasing the number of channels.

## 4. Conclusions

In summary, a water-modulated, continuous-flow droplet reactor has been developed for the scalable mass production of Ag NWs. Discrete droplets significantly enable the prevention of the back-flow or mixed-flow of the reaction solution, leading to good product uniformity and size control of nanosynthesis. Benefiting from the droplet reactor design, an optimized growth formula of Ag NWs is obtained, where the inner diameter of PTFE pipe (6 mm), the reaction residence time (60 min), and a trace of water (0.3%, *V/V*%) were determined. Finally, pilot-test equipment was designed and built accordingly, which could achieve a daily output at the kilogram scale. This work provides a practical solution for the large-scale production of Ag NWs, which is expected to promote the industrialization of Ag NWs for next-generation flexible displays with a high commercial value. Moreover, this flow droplet reactor could be extended to the scalable production of other nanomaterials.

## Figures and Tables

**Figure 1 nanomaterials-12-01018-f001:**
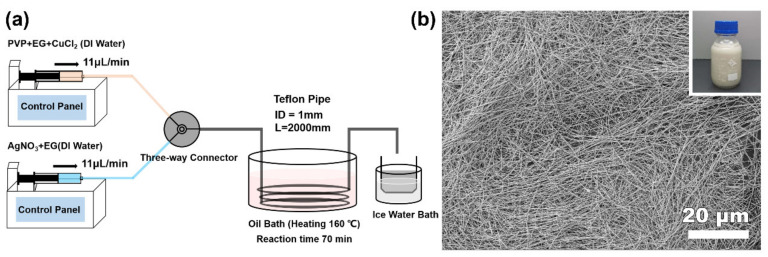
(**a**) Schematic illustration of a single-channel continuous-flow reactor for scalable production of Ag NWs. (**b**) Representative SEM image of Ag NWs obtained by such a continuous-flow reactor.

**Figure 2 nanomaterials-12-01018-f002:**
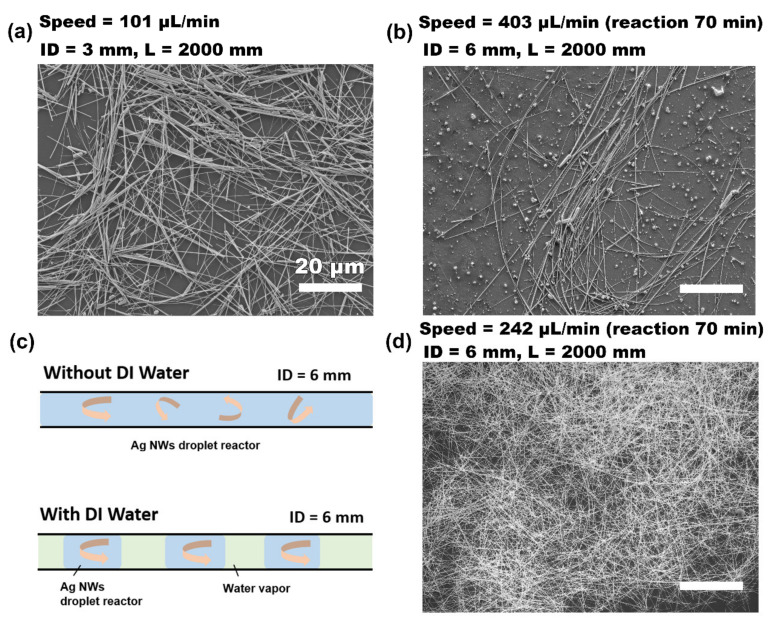
Diagrams of droplet flow-chemistry-processed Ag NWs against the inner diameters of PTEF pipe channel and water carrier phase. (**a**,**b**) SEM images of Ag NWs produced without DI water under pipe inner diameters of 3 and 6 mm, respectively. (**c**) The state of the fluid segment in the pipeline with or without deionized water. (**d**) SEM image of Ag NWs produced with DI water under a pipe inner diameter of 6 mm.

**Figure 3 nanomaterials-12-01018-f003:**
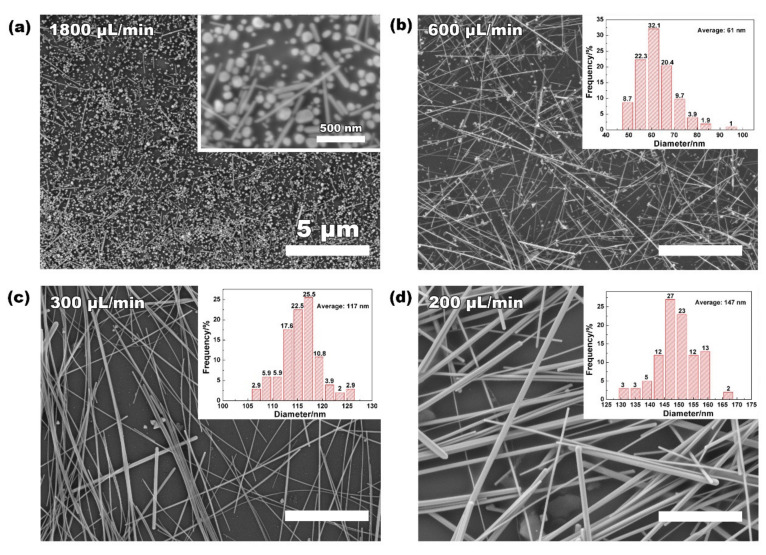
Flow-rate (reaction time)-dependent Ag NW growth. (**a**–**d**) Representative SEM images showing the morphology of Ag NWs grown at flow rates of 1800, 600, 300, and 200 uL/min, respectively. Inset shows the related size distribution of Ag NWs.

**Figure 4 nanomaterials-12-01018-f004:**
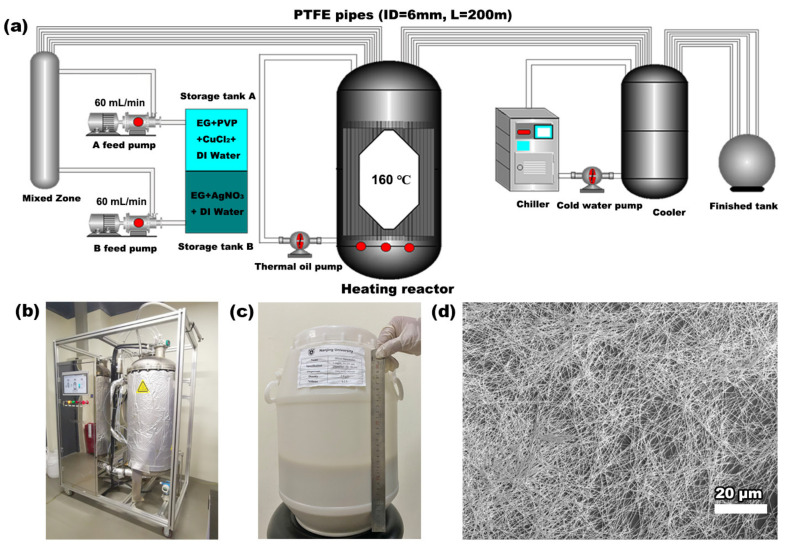
Pilot test of mass production of Ag NWs. (**a**) Schematic showing the configuration of the multi-channel droplet reactor equipment. (**b**) Photo of the overall pilot test reactor equipment. (**c**) Photo of Ag NWs dispersion product in a storage tank from one-hour production run. (**d**) SEM image of pilot-test-line-produced Ag NWs.

**Table 1 nanomaterials-12-01018-t001:** Effects of flow rate on aspect ratio of Ag NWs.

Flow Rate (uL/min)	Reaction Residence (min)	L (um)	D (nm)	Aspect–Ratio (L/D)
1800	10	*	*	*
600	30	28 ± 5	61 ± 10	459
300	60	47 ± 3	117 ± 10	402
200	90	51 ± 5	147 ± 43	347

## Data Availability

The data are available upon request from the corresponding author.

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
