# Peer review of "Scalable Production of High-Quality Silver Nanowires via Continuous-Flow Droplet Synthesis"

_nanomaterials, 2022, doi:10.3390/nano12061018_

Round 1
Reviewer 1 Report
I read the paper entitled "Scalable Production of High-Quality Silver Nanowires via Continuous-Flow Droplet Synthesis" that describes a reliable method for large scale synthesis of AgNWs. In the main, I believe that the paper would benefit from a clearer overview of the droplet reactors with their pros and cons, and this would then support the value of the solution proposed in this work. Also the reasons behind the shift from a continuous-flow reactor to a droplet one should be more clearly addressed, especially in the discussion of the experimental results (Pag.3 line131-133). Some specific comments are reported in the pdf file attached.

Reviewer 2 Report
In this paper the authors exploit the continuous-flow synthesis for scalable production of Ag NWs with high structural quality for mass production with industrial standard of kg/day. The analyses are accurate, the research concept is linear and clear and English writing is understandable, hence I recommend the publication on the work on the journal of Nanomaterials with minor revision and the following suggestions.
1. the introduction section is well written although I would suggest to add a few lines on the role of electrospunning techniques.
2.I am not sure the produces material are NWs, you should provide a reference on their definition, NWs are defined with diameter below 100 nm, the others are otherwise name microwires, and only some conditions lead to the formation of nanowires with average diameter of about 60 nm. Hence this aspect should be corrected along the main text.
3. The scalability of the process is impressive. Could you please provide an orientative estimation for mass production cost?
4. Silver is not stable after air exposure. Could you do some analysis on their purity and stability in time? Otherwise please add that you do not know or extended this aspect. Please add also some references for comparison.
Reviewer 3 Report
The manuscript by Yu et al. presents a continuous flow-drop process for synthesis of Ag nanowires where they provide an understanding of key mechanistic requirements, characterize the process in terms of key experimental parameters, and demonstrate its scalability. I am of the opinion that the manuscript is of high quality, interesting, well-presented, and addresses a topic of considerable importance. The quantities of nanowires produced is impressive. I, therefore, recommend it for publication but where the authors should first consider the following suggestions.
(1) One of the key metrics for a scalable process is that it shows a consistent product over time. While this may be the case for the nanowire process demonstrated, there is currently no evidence to this effect. This should be addressed.
(2) In the Materials section (lines 73-76) there is no mention of AgNO3.
(3) in the abstract it refers to trace water “originally present in the solvent” but this is never mentioned again in the manuscript. This needs some clarification.
(4) Figure 4a would be more intuitive if it were inverted since the process would then progress from left to right (i.e., put the “finish tank” near the right margin instead of the left).
(5) Some corrections to the English include (i) Line 14 should be “trace quantities of water”, (ii) line 78-79 – it is unclear what “respectively” is referring to –the same grammatical error is made throughout the manuscript almost every time the word “respectively” is used, (iii) line 104 should be “fed into the syringe”, (iv) line 118 “As aforementioned” should be removed, and (v) line 152 “no sufficient” should be “insufficient”.
